# Targeting Fear of Cancer Recurrence with Internet-Based Emotional Freedom Techniques (iEFT) and Mindfulness Meditation Intervention (iMMI) (BGOG-gyn1b/REMOTE)

**DOI:** 10.3390/brainsci15090900

**Published:** 2025-08-22

**Authors:** Laura Tack, Lore Mertens, Marte Vandeweyer, Fien Florin, Emma Pauwels, Thaïs Baert, Tom Boterberg, Christel Fontaine, Kurt Geldhof, Caroline Lamot, Christine Langenaeken, Jeroen Mebis, Laure-Anne Teuwen, Katherine Vandenborre, Heidi Van den Bulck, Florence Van Ryckeghem, Mohammad Najlah, Patricia Schofield, Philip R. Debruyne

**Affiliations:** 1Department of Medical Oncology, OECI-Designated Kortrijk Cancer Centre, General Hospital Groeninge, 8500 Kortrijk, Belgium; laura.tack@azgroeninge.be (L.T.); lore.mertens@azgroeninge.be (L.M.); marte.vandeweyer@azgroeninge.be (M.V.); 2Mindfulness Based Cognitive Therapist, 8500 Kortrijk, Belgium; 3Department of Medical Oncology, Leuven University Hospital, 3000 Leuven, Belgium; thais.baert@uzleuven.be; 4Department of Human Structure and Repair, Ghent University, 9000 Ghent, Belgium; tom.boterberg@uzgent.be; 5Department of Radiotherapy, Ghent University Hospital, 9000 Gent, Belgium; 6Department of Medical Oncology, Brussels University Hospital, 1090 Jette, Belgium; christel.fontaine@uzbrussel.be; 7Department of Medical Oncology, Jan Yperman Ziekenhuis, 8900 Ieper, Belgium; kurt.geldhof@yperman.net; 8Department of Medical Oncology, VITAZ, 9100 Sint-Niklaas, Belgium; caroline.lamot@vitaz.be; 9Department of Medical Oncology, AZ Klina, 2930 Brasschaat, Belgium; christine.langenaeken@klina.be; 10Department of Medical Oncology, Jessa Ziekenhuis, 3500 Hasselt, Belgium; jeroen.mebis@jessazh.be; 11Department of Oncology, Antwerp University Hospital, 2650 Edegem, Belgium; laure-anne.teuwen@uza.be; 12Department of Medical Oncology, AZ Vesalius, 3700 Tongeren, Belgium; katherine.vandenborre@azvesalius.be; 13Department of Medical Oncology, Imelda Ziekenhuis, 2820 Bonheiden, Belgium; heidi.van.den.bulck@imelda.be; 14Department of Medical Oncology, AZ Glorieux, 9600 Ronse, Belgium; florence.vanryckeghem@azglorieux.be; 15Medical Technology Research Centre (MTRC), School of Allied Health and Social Care, Faculty of Health, Medicine and Social Care, Anglia Ruskin University, Cambridge CB1 1PT, UK; mohammad.najlah@aru.ac.uk; 16School of Nursing and Midwifery, Faculty of Health, University of Plymouth, Plymouth PL4 8AA, UK; patricia.schofield@plymouth.ac.uk

**Keywords:** fear of cancer recurrence, emotional freedom techniques, mindfulness meditation, internet-based interventions

## Abstract

Approximately one-third of cancer survivors report a need for professional help to cope with Fear of Cancer Recurrence (FCR). In the REMOTE trial, we aim to investigate the efficacy of two internet-based mind-body techniques to address this currently unmet medical need. Cancer survivors, screened using the Cancer Worry Scale (CWS), are randomly assigned to one of the three study groups: (1) internet-based emotional freedom techniques (iEFT) (*n* = 113), (2) an active control condition internet-based mindfulness meditation intervention (iMMI) (*n* = 113), or (3) a wait-list control group (WLC) (*n* = 113). The interventions iEFT and iMMI are conducted remotely using Microsoft Teams (Microsoft, Redmond, WA), and participants have access to an online platform via the MyNexuzHealth application (nexuzhealth NV, Hasselt, Belgium). The primary endpoint is the level of FCR. Secondary outcomes are emotional distress and quality of life (QoL). If iEFT and/or iMMI appear to be effective in reducing FCR, they could be readily implemented in clinical practice.

## 1. Background

Recent advances in cancer treatment have substantially increased the number of cancer survivors. In 2022, the number of cancer survivors who were diagnosed in the past five years was estimated to be 53.5 million worldwide [1]. Despite the medical progress, mental well-being and quality of life (QoL) after cancer diagnosis and treatment remain poorly addressed in clinical practice [2].

Fear of Cancer Recurrence (FCR) is one of the most frequent psychological complaints reported by cancer survivors, with a prevalence ranging from 39% to 97% [3]. Research on long-term QoL consistently shows that this fear, worry, or concern about cancer returning or progressing is persistent and does not decline with time, although this might be expected [4,5,6,7]. Depending on its severity, FCR can profoundly impact both health and QoL [8].

Approximately one-third of cancer survivors report a need for professional help to cope with FCR [9,10,11]. Although the number of randomised controlled trials (RCTs) investigating FCR is increasing, evidence regarding the effectiveness of these interventions remains limited [9]. At present, there is no widely adopted, evidence-based approach to managing FCR.

The latest theoretical framework for FCR emphasises its multidimensional nature and underscores the role of cognitive processing and metacognition in the development and persistence of FCR [12]. According to this model, improving awareness of thoughts may be a therapeutic approach to alleviate worrisome and unhelpful thoughts that drive FCR [12]. This focus on cognitive awareness is a fundamental component of both Emotional Freedom Techniques (EFT) and mindfulness-based interventions (MBIs) [13,14].

Clinical EFT integrates elements from Cognitive Behavioural Therapy (CBT) and Prolonged Exposure therapy (PE) [15]. These include awareness building, imaginal exposure, cognitive reframing, pre-framing, and systematic desensitisation [15]. To this psychological component, the somatic component of acupressure or “tapping” is added. A meta-analysis by Church et al. demonstrated that integrating acupressure with established psychological interventions significantly enhances the overall therapeutic efficacy of EFT [16].

Mindfulness meditation involves cultivating moment-to-moment awareness of internal and external experiences with an accepting and open attitude [17]. Its beneficial effects on psychological well-being are well established [18]. A recent RCT illustrated the effectiveness of a mindfulness meditation intervention (MMI) to reduce anxiety and depression symptoms in breast cancer patients, which supports our choice for this method as an active control [19,20,21].

In the REMOTE trial (acronym derived from feaR rEcurrecce eMotional freedOm Techniques mindfulnEss), endorsed by the Belgian Society of Medical Oncology (BSMO) Cancer Survivorship Task Force and Belgian Gynaecological Oncology Group (BGOG; BGOG-Gyn1b), we opted for a tele-health or internet-based (i) approach to deliver the interventions: iEFT and iMMI. Online interventions offer greater accessibility, allowing individuals to engage at any time, in their own environment, with the option to remain anonymous. Consequently, tele-health has been established as a viable and effective modality [22,23,24]. Additionally, it holds significant potential to enhance patient empowerment.

This patient-centred, multicentre, pragmatic randomised controlled trial (RCT) aims to compare the efficacy of iEFT, the experimental intervention, and iMMI, the active control, with usual care (a waitlist), as an intervention to reduce FCR in cancer survivors. If proven to be efficacious to reduce FCR, iEFT and/or iMMI could be easily implemented in clinical practice.

## 2. Methods/Design

### 2.1. Aim and Objectives

The primary aim of this study is to examine whether two different interventions (iEFT and iMMI) can improve FCR experienced by cancer survivors, compared to a usual care wait-list control (WLC) group. To translate a statistically significant effect on FCR into a clinically significant change, we would need to detect a between-group difference in mean FCRI at T1 of 10 points (with T1 6 weeks after participation in the iEFT or iMMI programme, or 6 weeks waiting list; see Figure 1).

The secondary and tertiary objectives are listed in Table 1.

Our hypotheses are:iEFT improves FCR compared to the WLC group at T1iMMI (active control intervention) improves FCR compared to the WLC group at T1

### 2.2. Design

We designed a pragmatic multicentre RCT with three study arms and stratified random allocation. Participants in the WLC group receive care from the treatment centre as usual, and participants in the intervention groups also have access to the usual care in addition to participating in the 6-week iEFT or iMMI programme. Although both interventions require patient engagement, the “live attendance” is organised differently: the intervention format of the iMMI programme includes six weekly 2-h sessions, while the duration of the three individually guided iEFT sessions varies. A first iEFT session lasts about 90 min, the second 45 min, and a third 25 min. Time toxicity is included as one of the objectives (i.e., time spent per patient). The study evaluation will be done by comparing the within-group and between-group differences. The potential effect will be assessed over time on a series of patient-reported outcome measures (PROMs). Measurements will take place at baseline (T0; before randomisation), 6 weeks (T1; 6 weeks after T0 for the WLC and 6 weeks after the first iEFT or iMMI session for the intervention groups); 12 weeks (T2; 12 weeks after T0 for the WLC and 12 weeks after the first iEFT or 6 weeks after the last iMMI session for the intervention groups); 24 weeks (T3; 24 weeks after T0 for the WLC and 12 weeks after the last iEFT or iMMI session for the intervention groups). Participants are allowed to withdraw at any time. There will be no follow-up assessments of these drop-outs. The development of the study protocol followed the SPIRIT (Standard Protocol Items: Recommendations for Interventional Trials) guidelines [25]. The planned flow diagram of the REMOTE trial is presented in Figure 1. The protocol is reported according to the SPIRIT guidelines (Appendix A).

### 2.3. Eligibility Criteria

The eligibility criteria are listed in Table 2.

### 2.4. Setting and Recruitment Process

The recruitment takes place in twelve hospitals located throughout Flanders and Brussels, Belgium (Appendix A). Patients coping with FCR are identified by their treating oncologists or another healthcare professional involved throughout their disease trajectory. They can also be recruited through publicity, such as flyers in waiting rooms or projections on screens in the oncology units. Participation in the trial is also discussed with the healthcare professional (medical oncologist, GP, or psychologist), and this is explicitly mentioned in the ICF.

Potential candidates will receive a general outline of the study. When interested in participation, the assigned PI or his/her designee will complete the informed consent form (ICF) and screening procedure with the Cancer Worry Scale (CWS) 6-item version [26] (Appendix B). When eligible for participation (i.e., score of 10 or higher on the CWS; see Table 2), the local PI or his/her designee completes the electronic Case Report Form (eCRF), followed by the QuAno form (Quest Anonymous by NexuzHealth; nexuzhealth NV, Hasselt, Belgium) and allocates the participant to one of the three groups in REDCap (Vanderbilt, Nashville, TN) [27]. Upon completion of the local procedure, the central project coordinator (L.T., L.M., M.V.) from the coordinating site az groeninge (Kortrijk, Belgium), follows up on the inclusion. It’s the central project coordinator’s responsibility to contact each participant by phone to explain further details regarding study participation and address any questions.

### 2.5. Sample Size Calculation

The sample size calculation takes into account the principle of minimization for randomization, with participants randomized into either the iEFT group, iMMI group, or WLC group. In order to achieve at least 80% power to detect a between-group difference in mean FCRI at T1 of 10 points using an independent samples *t*-test at the two-sided 2.5% significance level (two experimental groups are compared against a single control), a total sample size of 444 patients (148 in each group) is needed, assuming a common standard deviation for FCRI in all groups of 27.9 points.

When taking into account baseline FCRI as a covariate in the regression model, the calculated sample size can be reduced by a design factor equal to (1 – r^2^) [28]. Assuming a correlation between baseline and follow-up FCRI of 0.6, the sample size reduces to 288 patients in total. This sample size has been increased to 339 patients in total to allow for an expected drop-out rate of 15%. This means we will include 113 patients in each group: iEFT (intervention), iMMI (active control), and the WLC group.

### 2.6. Randomisation

Participants are randomised (1:1:1) to the control group (WLC) or intervention groups (iEFT or iMMI) stratified by age (< or ≥50 years), chemotherapy (yes or no), and gender (male or female). The stratified randomisation will be performed at the local sites by the site PI or his/her designee using the randomisation module of REDCap. REDCap is a secure, web-based application designed to support data capture for research studies. The REDCap software (REDCap 14.3.0) will randomise patients based on the statistical method with random permuted blocks that will be created with a computer random number generator with variable sizes to avoid the treatment allocation being predicted [27].

After randomisation, participants will be unblinded to group assignment, as the interventions do not allow for blinding. For analysis, assessor(s) and statistician(s) will be blinded.

### 2.7. Interventions

The intervention will be remotely delivered through Microsoft Teams, as the Microsoft Office 365 package is sufficiently covered for all patient data privacy requirements by the coordinating site az groeninge (MS Teams complies with the provisions of the HIPAA Security Rule and GDPR privacy and security law). Participants will all be well-informed on the use of MS Teams before signing the ICF.

#### 2.7.1. Intervention Group—iEFT

Participants allocated to the iEFT group follow three iEFT sessions, individually guided by a clinical psychologist trained in EFT (EFT level 1 certificate acknowledged by EFT International [EFTi, Tiverton, United Kingdom]). During the trajectory, there is an in-depth initial session (±90 min) and two additional sessions (lasting ±45 and ±25 min, respectively). The duration of each session is stored in the electronic health record (i.e., measurement of time toxicity; Table 1). All sessions are guided by the iEFT practitioner on a 1:1 level.

During the first iEFT session, participants will be taught how to apply EFT and how it can be used as an application for mood and physical side effects. They will receive a digital information brochure including a section that illustrates the application of the tapping protocol. This visualisation of tapping points (Figure 2) and self-measurement scale serves as a tool to support and encourage home practice. Patients may indicate in the MyNexuzHealth app whether and when they have applied EFT. For participants who prefer a paper version of the ‘EFT diary’, it is allowed to print out the forms at home. This (digital) notebook containing a structured self-work outline will be consulted during the follow-up iEFT sessions to evaluate the previous weeks and one’s evolution throughout the iEFT trajectory.

Each session will give tools to the participants to facilitate their practice of EFT at home through formal EFT practice (together with the iEFT practitioner) as well as through guidance in the informal use of EFT during daily life. During each session, the iEFT practitioner will assist the participants in learning this practice and will regularly assess how the participants are doing with their EFT practice. An overview of the iEFT programme is presented in Appendix C.

#### 2.7.2. Active Control Group—iMMI

The specific curriculum for this 2 h/week, 6-week programme will be provided to cancer survivors who are assigned to this group as part of the randomisation. Each session will provide structured training and exercises in mindfulness, and cancer survivors will be given tools to facilitate their practice of these techniques at home through formal meditation practice as well as through guidance in the informal use of mindfulness during daily life. During each session, the instructor will assist the participants in learning this practice and will regularly assess how the participants are doing with their mindfulness practice. All sessions are guided by a clinical psychologist who is a certified mindfulness trainer. A detailed training manual is presented in Appendix C.

#### 2.7.3. Wait-List Control Group

The participants assigned to this group after randomisation will not receive any specific intervention until they have completed their T2 research assessment (12 weeks after the assignment to the WLC group). Hereafter, participants will be offered participation in the iEFT or iMMI programme after they have completed the assessments. The participants allocated to the wait-list may thus choose their preferred programme.

### 2.8. Measures

An overview of measures is given in Table 3 and Appendix A.

Participants may consent to deliver a hair sample at T0 and T1 to explore the effect of both interventions on chronic biological stress measured as hair cortisol concentration. For the analysis of the levels of cortisol in the hair samples, a separate protocol will be developed, and approval will need to be obtained from the ECs of all participating sites in the REMOTE trial. Analysis would take place in collaboration with the ARU biomarker laboratory (Cambridge, UK) [30]. The laboratory holds a Human Tissue Authority license and is the only Salimetrics (Carlsbad, CA, USA) approved centre of excellence for testing in the UK.

### 2.9. Analysis Plan

Primary outcome measures:The difference in mean scores from the baseline on the FCRI between the iEFT group and the WLC group at T1;The difference in mean scores from the baseline on the FCRI between the iMMI group and the WLC group at T1;

Secondary outcomes measures:Changes in emotional distress, QoL, and health status;The difference in mean scores from the baseline on the FCRI between the iMMI group and the WLC group;

Tertiary outcomes measures:Changes in the cortisol biomarker

### 2.10. Data Analysis

All planned analyses will be performed according to the intention-to-treat principle.

Descriptive statistics will be used to describe demographic and clinical characteristics.

Clinical and psychosocial outcomes will be compared between groups using linear mixed models. Linear mixed-effects models will assess whether the improvement in FCRI scores and secondary PROMs is significantly different over time. Models will explore time x group as an interaction effect and will include the stratification variables (age, chemotherapy, and gender). Estimated marginal means with 95% confidence interval (CI) will be calculated for each time point for both groups. A 5% significance level will be used.

Within-group differences will be calculated by pairwise comparisons, enabling us to detect differences over time within the three groups. Between-group differences in the primary outcome at the primary endpoint T1 will be further examined with pairwise comparisons with T0 as the reference time point. These analyses will be repeated for the secondary PROMS: distress, QoL, and health status.

### 2.11. Data Management

Data collection will be conducted using an electronic data capture (EDC) system (Vanderbilt University, TN), called REDCap. All participating study sites will be granted access to REDCap, with access controls implemented via IP address restrictions to ensure secure and authorised use. Data entry and any corrections will be restricted to site personnel who have completed GCP training and have been formally authorised by the principal investigator.

Upon enrolment in the study, each participant will be assigned a unique study number or “participant ID” within the REDCap system, ensuring pseudonymisation. In all documentation to which the coordinating centre, sponsor, or Chief Investigator (CI) has access, this number will solely identify participants. The list that matches these participant IDs to the participants’ names is safeguarded by the local site. Any personal identifiers will not be recorded in the study database (REDCap).

All entries in the eCRFs must fully align with the corresponding source data. In instances where data are unavailable or unknown, this must be explicitly documented within the eCRF. The eCRFs are designed to capture all variables specified by the study protocol (see Appendix A). Data quality and consistency will be systematically reviewed by qualified personnel, including clinical research monitors and data managers, with any discrepancies subject to query and resolution in accordance with standard data management procedures.

Users are limited to viewing data pertaining only to participants enrolled at their respective study sites. All user activity within the REDCap system is automatically recorded and traceable via comprehensive audit trails and log files, ensuring transparency and accountability.

### 2.12. Ethics Approval

The finalized and operational study protocol was approved by the Medical Ethics Committee of all participating centres, and final approval was obtained by the Central Medical Ethics Committee of UZ Brussel/VUB on 3 July 2024 (23435_REMOTE). This study was registered with ClinicalTirals.gov (https://clinicaltrials.gov/study/NCT06175208) on 8 December 2023 [31].

## 3. Results

The initiation visits for the REMOTE study took place between September and December 2024 in all twelve participating centres. The first patient was randomised on 16 October 2024. The number of participants currently stands at 82 of the target 339 on 31 July 2025.

## 4. Discussion

The aim of this study is to examine the efficacy of iEFT and iMMI in alleviating FCR, one of the most frequently reported psychosocial problems by cancer survivors. Based on the outcomes of the former EMOTICON trial [32], it is expected that iEFT may control and diminish FCR experienced by cancer survivors. The integration of exposure therapy, cognitive framing, and acupressure [15] renders EFT a straightforward yet effective tool to manage negative thoughts. Furthermore, it is a patient-friendly intervention that can be easily delivered remotely.

For the active control arm in the study, iMMI, it is expected that this intervention arm will also be able to reduce FCR. In addition, this intervention group may offer novel insights into the impact of this particular mindfulness approach on FCR and on the overall psychological well-being of cancer survivors.

One of the potential limitations related to the type of intervention and topic of the trial assumes a potential selection bias for white, highly educated, female participants. However, the research group consciously decided to include all cancer types as all cancer survivors may cope with (severe) levels of FCR. We have anticipated by including the biggest risk factors for selection bias (age, gender), which are also related to higher levels of FCR, as stratification variables. Hereby, the intervention arms should all contain equal distribution of younger/older and male/female participants. Race and ethnicity are not taken into consideration, and this may potentially be a topic for the final discussion together with the general analysis of the demographic data.

Another limitation could be the entire online format, which increases the risk for drop-out. We may anticipate this by keeping close contact with local trial coordinators in the first place, stressing the importance of the eligibility criteria, but even more the importance of the motivation of participants. Next, it is the central coordinator’s responsibility to follow up on all participants and the response to the online questionnaires via MyNexuzHealth.

Positive outcomes may promote the integration of iEFT and iMMI into clinical practice. Therefore, the time toxicity of both interventions is measured as this may significantly impact the feasibility of implementation.

## 5. Conclusions

In conclusion, this pragmatic RCT is of great importance given the considerable need and demand from cancer survivors for effective strategies to manage FCR [9]. Identifying a low-threshold, cost-effective intervention could have a substantial impact on the survivorship trajectory and enhance the overall experience of this expanding population. Moreover, the internet-based delivery of the interventions may facilitate an effective and scalable implementation in routine clinical practice, the ultimate objective of this research. The involvement of the national scientific societies, such as the BGOG and BSMO, will be helpful in facilitating this potential implementation in clinical practice. In addition, their involvement can also help to promote integration into international guidelines, so that every patient coping with FCR can have access to this form of supportive cancer care.

## 6. Trial Status

The protocol number is 2345_REMOTE and concerns version 2.0 (dd. 1 July 2024). Recruitment started in October 2024 and is projected to be finished in March 2027.

## Figures and Tables

**Figure 1 brainsci-15-00900-f001:**
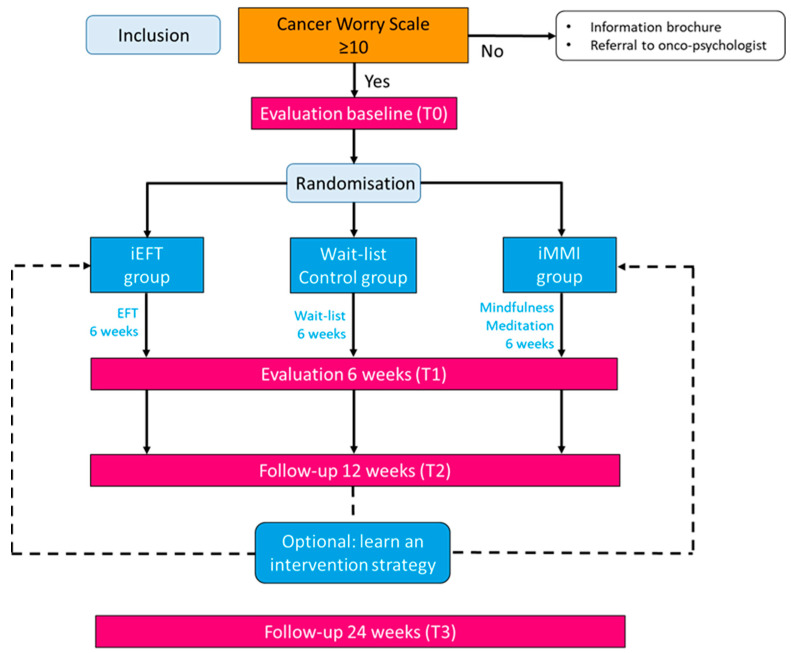
Diagram of planned study flow.

**Figure 2 brainsci-15-00900-f002:**
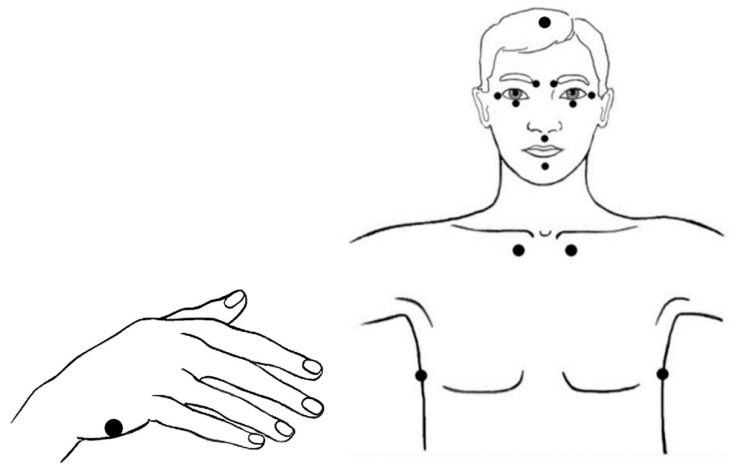
Head, torso, and hand acupressure points involved in the application of EFT (Figure adapted from Church et al. [29]). The process starts with a ‘Setup Statement’ while performing acupoint tapping on the side of the hand. For the remainder of the process, patients gently tap with two fingers on head and torso points while repeating the ‘Reminder Phrase’.

**Table 1 brainsci-15-00900-t001:** Secondary and tertiary objectives.

Secondary objectives
• Explore whether iEFT is more effective than iMMI for improving FCR
• Compare the efficacy of the two interventions relative to a WLC group on distress, QoL and health status
• Explore whether iEFT is more effective than iMMI (active control condition) for improving distress, QoL and health status
• Prevalence and predictors of FCR in cancer survivors
• The magnitude of the interventions to reduce FCR in cancer survivors
• The level of preference of the WLC group to (optionally) complete a course of iEFT or iMMI
• The level of efficacy of both interventions between younger and older cancer survivors (< or ≥50 years)
• The level of the efficacy of both interventions between cancer survivors participating within 1 year after ending curative therapy and cancer survivors participating ≥1 year after ending curative therapy
• The time spent by iEFT and iMMI practitioners (time toxicity per patient)
Tertiary objective *
• The effect of iEFT and iMMI on chronic biological stress measured as hair cortisol concentration.

* This objective will only be completed when the primary objective is positive.

**Table 2 brainsci-15-00900-t002:** Inclusion and exclusion criteria.

Inclusion criteria
• Minimum age of 18 years at the time of enrolment
• Dutch as main language
• Histologically confirmed diagnosis of a solid cancer or haematological malignancy
• Curative surgery, chemotherapy, radiation therapy, immunotherapy, hormone therapy, targeted drug therapy or a combination of these was completed at least 2 months ago and no longer than 5 years ago at the time of enrolment. *
• Expected life expectancy of at least 5 years
• Disease-free at the time of enrolment, as defined by the absence of somatic disease activity parameters
• High FCR based on the Cancer Worry Scale 6-item version [26] (cut-off ≥ 10)
• Sufficient mental and physical functional status
Exclusion criteria
• Treatment with palliative intent
• Mental deterioration
• Organic brain syndrome (concussion without neurological symptoms and negative imaging is allowed)
• Alcohol or drug dependent
• Serious or chronic medical, neurologic or psychiatric condition that contributes to substantial physical or emotional disability that would detract from participating in either of the intervention programmes or from the measurement of intervention outcomes; a prior diagnosis of a depressive, anxiety or adjustment disorder is allowed
• Commitment to the intervention schedule is not possible
• Active practice of EFT or mindfulness (based) meditation

* Patients under current treatment for a depression or anxiety disorder are allowed to enrol, provided their depression or anxiety disorder is stable.

**Table 3 brainsci-15-00900-t003:** Overview of measures and corresponding measurement time points ^1^.

Measure	T0	T1	T2	T3
**Sociodemographic data ^2^**	X			
**Medical data ^2^**				
**PROMs ^3^**				
Fear of Cancer Recurrence Inventory ^4^	X	X	X	X
Distress thermometer	X	X	X	X
38-item Problem List	X	X	X	X
EORTC QLQ-SURV100 ^5^	X	X	X	X
EuroQol EQ-5D-5L	X	X	X	X
**Biomarker cortisol** *	X	X		

Abbreviations: PROMs: Patient Reported Outcomes Measures; EORTC: European Organisation for Research and Treatment of Cancer; QLQ: Quality of Life Questionnaire; SURV100: survivorship questionnaire with 100 items; EuroQol: European Quality of Life. ^1^ T0: baseline (i.e., before randomisation); T1: 6 weeks after start of the iEFT or iMMI programme or 6 weeks after randomisation to the wait-list; T2: 6 weeks after end of the iEFT or iMMI programme or 12 weeks after randomisation to the wait-list; T3: 24 weeks after start of the iEFT or iMMI programme or 24 weeks after randomisation to the wait-list. ^2^ Details can be found in Appendix A. ^3^ Details can be found in Appendix A. ^4^ Details can be found in Appendix D. ^5^ Additionally, item 13, 14, 15, 16 and 17 of the EORTC QLQ-C30 have been added. * Optional, needs to be indicated in the Informed Consent Form (ICF).

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
