# Peer review of "Targeting Fear of Cancer Recurrence with Internet-Based Emotional Freedom Techniques (iEFT) and Mindfulness Meditation Intervention (iMMI) (BGOG-gyn1b/REMOTE)"

_brainsci, 2025, doi:10.3390/brainsci15090900_

Round 1

Reviewer 1 Report

Comments and Suggestions for Authors

Comment 1. The title is too long: “BGOG-gyn1b/REMOTE: Targeting Fear of Cancer Recurrence with internet-based Emotional Freedom Techniques (iEFT) and internet-based Mindfulness Meditation Intervention (iMMI): A Study Protocol for a Multicentre, Randomized Controlled Clinical Trial.” Please consider shortening it while maintaining the core meaning of the manuscript.

Comment 2. The abstract is overly complex and lacks a logical flow. It should provide a general overview that attracts readers’ interest, but in this case, the project’s objective is unclear, there is no background context, and the contribution of the study remains vague. There are also several grammatical and spelling errors that need correction.

Comment 3. The mention of progress since the COVID-19 pandemic appears unnecessary, especially as it is not a central aspect of the study. I recommend restructuring the last three paragraphs of the introduction. Additionally, there are some uncited statements in this section that must be properly referenced.

Comment 4. In the methodology section, I noticed the inclusion of objectives, which should not be placed in the introduction. Also, avoid listing a large number of items. Consider summarizing these points in prose or presenting them in a table for better readability.

Comment 5. Is the methodology proposed by the research group, or is it based on an existing framework? This needs to be clearly stated and appropriately cited.

Comment 6. The manuscript mentions a tertiary objective regarding the effect of both interventions on chronic biological stress measured by hair cortisol levels. However, there is no clear mention of a secondary objective. It is important to clarify whether the listed objectives are general or specific and present them in a logical order.

Comment 7. “Time toxicity” is also mentioned as an objective, but again, the list of objectives is unclear and inconsistent.

Comment 8. The methodology section is excessively long. Consider summarizing the core elements in the manuscript and moving the more detailed content to the supplementary material.

Comment 9. The timeline presented in the text is confusing and repetitive, which makes reading difficult. Concepts and ideas are reiterated without adding clarity. Please revise this section for consistency and conciseness.

Comment 10. In the diagram, terms like “Evaluation 6 weeks” and “Follow up 12 and 24 weeks” are not clearly explained or aligned with the methodology section. Furthermore, elements such as “Inclusion” and “Follow up” appear disconnected from the rest of the flowchart. Please revise to ensure coherence.

Comment 11. Table 1 should be moved to the supplementary materials section.

Comment 12. The inclusion criterion of a minimum age of 18 years is stated, but no upper age limit is provided. Please clarify the age range being evaluated, as psychosocial issues vary significantly across different age groups.

Comment 13. It is unclear why a confirmed tumor diagnosis is required as an inclusion criterion if the study is supposed to focus on cancer survivors. Please explain this decision.

Comment 14. The methodology section should be entirely rewritten. It is difficult to read due to excessive length, redundant explanations, and too many diagrams. These materials could be included as supplementary files instead.

Comment 15. The results section is missing.

Comment 16. The discussion is too brief considering the complexity of the study. It should be significantly expanded.

Comment 17. Although the article addresses an important and relevant topic, it currently lacks the minimum structure and quality required for acceptance. Key sections are missing, there is redundancy in phrasing, and the manuscript does not follow MDPI’s formatting guidelines or citation style. I strongly recommend that the authors consult the “Instructions for Authors” section on the journal’s website to revise and improve the manuscript accordingly.

Comments on the Quality of English Language

The manuscript should be submitted to professional English language editing prior to further review.

Author Response

We would like to thank the reviewer for the time to read the manuscript and the thoughtful suggestions and comments that have been made. Please find attached the point-by-point reply.

Kind regards,

Dr. Laura Tack, on behalf of Professor Philip Debruyne

Reviewer 2 Report

Comments and Suggestions for Authors

In this article, titled “BGOG-gyn1b/REMOTE: Targeting Fear of Cancer Recurrence 2 with internet-based Emotional Freedom Techniques (iEFT) and 3 internet-based Mindfulness Meditation Intervention (iMMI): A 4 Study Protocol for A Multicentre, Randomized Controlled Clinical Trial,” the author explores the clinical advantages of using internet-based iEFT and iMMI methodology in targeting fear of cancer recurrence (FCR) compared to waiting list control groups, highlighting a detailed protocol in improving therapeutic outcomes. The topic is both timely and of clear clinical importance. However, before this manuscript can be advanced for further review, I recommend the following revisions to enhance its clarity, rigour, and broader relevance:

  1. The manuscript title is overly long and could be revised for greater conciseness and impact. A more reader-friendly title would enhance the manuscript’s appeal and readability.
  2. The authors mention three study groups, each consisting of 113 participants, including the wait-list control group. It is unclear whether this number was randomly determined or intentionally equal across groups. If the latter, the authors should clarify how they ensured demographic diversity (e.g., age, sex, race, ethnicity) while maintaining equal group sizes. This is essential for the generalizability of the findings.
  3. In current clinical settings, mindfulness-based practices and physical activities are commonly used to manage FCR. The authors should elaborate on how the proposed iEFT and iMMI approaches differ from, or improve upon, existing standard interventions. A clearer explanation of the novelty and added value of these internet-based methods would strengthen the rationale for the study.
  4. The manuscript would benefit from thorough language editing to correct grammatical issues and improve overall clarity and impact. Engaging a professional editor or utilizing a language editing service is recommended to enhance the manuscript’s readability and professionalism.

Comments on the Quality of English Language

Engaging a professional editor or utilizing a language editing service is recommended to enhance the manuscript’s readability and professionalism.

Author Response

(The authors gave the same response as above.)

Reviewer 3 Report

Comments and Suggestions for Authors

The abstract addresses a highly relevant and clinically valuable topic, namely the management of fear of cancer recurrence in survivors, a common concern that is often overlooked. I find it particularly positive that a methodologically sound design was used, with three well-defined arms and the use of internet-based mind-body interventions, which are increasingly necessary and in line with current trends in digital health. I also commend the use of validated instruments such as the Cancer Worry Scale (CWS) and the Fear of Cancer Recurrence Inventory (FCRI), which lend rigour to the evaluation of results. In terms of the structure of the abstract, it includes the essential elements (context, methodology, expected results and implications) and provides a general description of the study design. However, there are aspects of the writing that could be improved to achieve greater clarity and fluency. Some sentences are long or somewhat convoluted, making them difficult to read. For example, the expression ‘take off of the WLC randomisation’ is not entirely clear and could be replaced by something more direct, such as ‘assignment to the control group on the waiting list’. Likewise, details about the duration and frequency of the sessions could be organised more clearly to facilitate understanding. I also suggest avoiding unnecessary repetition, such as the repeated use of ‘internet-based’ in nearby sentences, and clarifying certain vague terms, such as ‘emotional distress’, indicating whether specific measures will be used to assess it.

The final lines mention the possible clinical impact of the interventions, which is very relevant, but I think the conclusion could be further strengthened by better explaining how these results could be translated into clinical practice or inspire future research. Overall, this is a well-focused abstract and a study with great potential, which with a little editorial revision would gain in clarity, consistency and precision.

The introduction presents a solid and well-documented foundation that clearly frames the clinical need addressed by the study. I would like to highlight the choice of topic—fear of cancer recurrence (FCR)—which remains a real and often underestimated concern in post-treatment care. The initial contextualisation of the increase in the number of cancer survivors and the persistence of LFRC over the long term is highly relevant and supported by up-to-date figures and references. This helps to clearly justify the purpose of the trial.

I also appreciate the inclusion of the most recent theoretical model on FCR and its link to cognitive processing and metacognitions, as it provides a coherent conceptual framework for the use of interventions such as EFT and mindfulness. The description of both approaches (including their basis in CBT and PE, in the case of EFT, and the proven positive effects of mindfulness) is well argued and supports the choice of interventions.

However, I suggest some adjustments to improve the flow and readability of the text. For example, some sentences could be simplified or reorganised to avoid repetition and make reading easier. In particular, the section on the use of telemedicine could benefit from more direct wording, as it currently feels somewhat fragmented. It would also be useful to make a smoother transition between the theoretical rationale and the presentation of the study objective, as this appears somewhat abrupt at the end of the text. I would also recommend clarifying from the outset that this is a pragmatic multicentre trial, rather than leaving this detail until the end. This would reinforce the applied and realistic nature of the study. Finally, although it is mentioned that the trial is supported by national scientific societies, it could be briefly added why this support is relevant to the validity or applicability of the results.

I believe that the methods section is well structured and provides a clear basis for understanding the study design. I think it was the right decision to opt for a three-arm randomised clinical trial, as this allows not only the efficacy of the two active interventions to be evaluated, but also for them to be compared with a control group on a waiting list. I also consider that the use of digital platforms such as Microsoft Teams and MyNexuzHealth is appropriate in the current context of telemedicine and makes the interventions accessible. I appreciate the details provided on key elements such as the number of participants per group, the duration and frequency of the sessions, and the assessment times. Furthermore, the use of validated instruments (such as the CWS for initial screening and the FCRI as the primary outcome measure) reinforces the methodological robustness. That said, I believe that some aspects could be clarified further. For example, I would like to see a more specific description of the inclusion and exclusion criteria for participants, as well as the cut-off point used in the CWS to consider a person eligible. I would also find it useful to know if there will be any additional clinical verification to confirm the level of CRF. I also consider it important to indicate the randomisation procedure: what type of sequence will be used, how concealment will be ensured, and whether there will be any type of blinding (e.g. for those analysing the data). Although I know that these details are sometimes elaborated further in the full protocol, a brief mention in the methodological summary would help to reinforce the transparency of the design. Finally, I see that the assessment time points are mentioned, but it would be advisable to add a brief reference to the type of statistical analysis that is planned to be used (even if only in general terms), such as intention-to-treat analysis or repeated measures models.

The discussion section clearly addresses the study objectives and the potential clinical value of the proposed interventions (iEFT and iMMI). I think it is positive that the previous experience of the EMOTICON trial has been revisited to support the expectation of iEFT's effectiveness, and that the accessibility and low cost of both interventions are highlighted, making them potentially viable for implementation in routine clinical practice. I also consider it appropriate to mention the patient-centred approach and to take into account aspects such as the ‘time toxicity’ of interventions, a relevant factor when considering transferring interventions to the real world. I note that the text also rightly introduces the importance of exploring not only the efficacy on CRF, but also the possible broader impact on the overall psychological well-being of survivors. This provides a more comprehensive view of the expected benefit. That said, I believe that this section could be enriched with a slightly more critical reflection on the limitations of the study (e.g., possible selection biases, limitations of the online format or follow-up), as well as a broader discussion of how the results could influence future research or long-term care policies in oncology. I also note that a final section on conclusions has not been included. I believe that a brief, clearly separated final paragraph would help reinforce the study's central message and make its specific contribution clearer. This could summarise the expected findings, their potential applicability and the importance of continuing research in this area.Overall, this is a well-focused discussion with an optimistic and realistic message about the potential for improving care for cancer survivors. With minor adjustments and an explicit conclusion, this section would gain even more strength and coherence within the article as a whole.

As for references, more than 40 references have been used, most of which are current.

In summary, this study addresses a highly relevant and necessary topic in cancer survivor care, namely the management of fear of recurrence, a concern that is often overlooked. The study design is robust and up-to-date, incorporating accessible, evidence-based digital interventions, which have great potential to positively impact patients' quality of life. With some adjustments to improve the clarity and cohesion of the text, this manuscript could make a valuable contribution to both the scientific literature and clinical practice. I hope the authors will consider these recommendations to enhance the scope and applicability of their research, which undoubtedly represents an important advance in comprehensive support for those facing this complex experience.

Author Response

(The authors gave the same response as above.)

Round 2

Reviewer 1 Report

Comments and Suggestions for Authors

The authors have adequately addressed nearly all of the previous observations, and the manuscript has significantly improved in both clarity and content. I consider the article suitable for acceptance. However, a final review of the references is recommended to ensure that all citations conform to the journal’s formatting standards.

Author Response

We would like to thank the reviewer for this positive feedback. The references have been reviewed according to the journal’s formatting standards.

Reviewer 2 Report

Comments and Suggestions for Authors

In the revised manuscript, the author has attempted to implement several changes aimed at improving clarity and scientific rigor. However, these revisions remain insufficient and, in some instances, have introduced additional confusion due to a lack of logical flow, as outlined below:

  1. Several sections, including Randomization (2.6), Interventions (2.7), Active Control Group (2.7.2), and Waitlist Control Group (2.7.3) read more like conceptual plans rather than components of a finalized and operational study protocol. It remains unclear whether any experimental procedures have actually been conducted or if these are proposed activities for future implementation. If these elements are merely intended for future execution, the author should clarify their inclusion in the current version of the protocol and explain their relevance at this stage.
  2. Although the study design, along with its aims and objectives, is reasonably well-presented, the sections on Measures, Data Analysis, and Data Management are poorly organized, ambiguous, and in some places internally inconsistent. As currently written, these sections appear hypothetical rather than based on completed work. If no data has been collected yet, it is important to explain the rationale for drawing conclusions or defining objectives in their current form. Substantial revision is necessary to improve clarity, consistency, and alignment with the study’s stated goals.
  3. The Results section is underdeveloped and lacks clarity. It currently reads more like an extension of the methods rather than a presentation of findings. This section should be revised to present and interpret the actual data outcomes.
  4. The Conclusion also needs refinement, particularly in sentence construction. For example, the statement “The involvement of the national scientific… may facilitate integration in international guidelines” is vague and difficult to interpret. This should be rephrased to convey a more precise and meaningful message.

Author Response

Response to Reviewer 2

Comments and Suggestions for Authors

In the revised manuscript, the author has attempted to implement several changes aimed at improving clarity and scientific rigor. However, these revisions remain insufficient and, in some instances, have introduced additional confusion due to a lack of logical flow, as outlined below:

  1. Several sections, including Randomization (2.6), Interventions (2.7), Active Control Group (2.7.2), and Waitlist Control Group (2.7.3) read more like conceptual plans rather than components of a finalized and operational study protocol. It remains unclear whether any experimental procedures have actually been conducted or if these are proposed activities for future implementation. If these elements are merely intended for future execution, the author should clarify their inclusion in the current version of the protocol and explain their relevance at this stage.

We would like to thank the reviewer for this remark. As suggested, we have clarified in the methods section that the finalized and operational study protocol was approved by the Ethics Committee on 3 July 2024 (p. 12, line 358). In the results section, we have mentioned that all twelve centres have been initiated between September and December 2024 and the data that the first patient has been randomised (p. 12, lines 365-367). On this date, 82 patients have been randomised of the target of 339 (p. 12, line 368).

  1. Although the study design, along with its aims and objectives, is reasonably well-presented, the sections on MeasuresData Analysis, and Data Management are poorly organized, ambiguous, and in some places internally inconsistent. As currently written, these sections appear hypothetical rather than based on completed work. If no data has been collected yet, it is important to explain the rationale for drawing conclusions or defining objectives in their current form. Substantial revision is necessary to improve clarity, consistency, and alignment with the study’s stated goals.

The sections on MeasuresData Analysis, and Data Management  have been revised accordingly (p. 10-12). We would like to clarify that no data have been analysed as the accrual is currently ongoing. This has been clarified in the results section (p. 12, lines 365-368).

  1. The Results section is underdeveloped and lacks clarity. It currently reads more like an extension of the methods rather than a presentation of findings. This section should be revised to present and interpret the actual data outcomes.

The inclusion of a results section was requested by another reviewer. However, as the REMOTE study started the recruitment period in October 2024, and no analysis have taken place, we cannot elaborate this section further with data outcomes. This has been further clarified (p. 12, line 358 and lines 365-368).

  1. The Conclusion also needs refinement, particularly in sentence construction. For example, the statement “The involvement of the national scientific… may facilitate integration in international guidelines” is vague and difficult to interpret. This should be rephrased to convey a more precise and meaningful message.

We would like to thank the reviewer for this feedback. Changes have been made accordingly  (p. 13, lines 407-412).